# Intrinsically Disordered Synthetic Polymers in Biomedical Applications

**DOI:** 10.3390/polym15102406

**Published:** 2023-05-22

**Authors:** Elif Yuce-Erarslan, Abraham (Avi) J. Domb, Haytam Kasem, Vladimir N. Uversky, Orkid Coskuner-Weber

**Affiliations:** 1Chemical Engineering, Istanbul University-Cerrahpasa, Avcilar, Istanbul 34320, Turkey; 2School of Pharmacy, Faculty of Medicine, Hebrew University of Jerusalem, Jerusalem 91120, Israel; 3Azrieli College of Engineering, 26 Ya’akov Schreiboim Street, Jerusalem 9103501, Israel; 4Department of Molecular Medicine, Morsani College of Medicine, University of South Florida, Tampa, FL 33612, USA; 5Molecular Biotechnology, Turkish-German University, Sahinkaya Caddesi, No. 106, Beykoz, Istanbul 34820, Turkey

**Keywords:** bioinspired polymers, intrinsically disordered polymers, drug delivery, organ transplantation, artificial organs, immune compatibility

## Abstract

In biology and medicine, intrinsically disordered synthetic polymers bio-mimicking intrinsically disordered proteins, which lack stable three-dimensional structures, possess high structural/conformational flexibility. They are prone to self-organization and can be extremely useful in various biomedical applications. Among such applications, intrinsically disordered synthetic polymers can have potential usage in drug delivery, organ transplantation, artificial organ design, and immune compatibility. The designing of new syntheses and characterization mechanisms is currently required to provide the lacking intrinsically disordered synthetic polymers for biomedical applications bio-mimicked using intrinsically disordered proteins. Here, we present our strategies for designing intrinsically disordered synthetic polymers for biomedical applications based on bio-mimicking intrinsically disordered proteins.

## 1. Introduction

In the last 10 years, polymers have been highly preferred in biomedical areas due to their relatively easy modification processes and other important features (Table 1). Specifically, polymers are required in drug delivery, organ transplantation, artificial organ design, and immune compatibility. Polymers play a critical role in drug delivery because they can be designed and synthesized to have specific chemical and physical properties that allow them to be used for different drug delivery applications [1,2,3,4,5]. Some of the roles that polymers play in drug delivery include: (i) Controlled release: Polymers can also be designed to encapsulate drugs via covalent bonds to release drugs slowly over an extended period of time, providing sustained therapeutic effects and reducing the need for frequent dosing [6,7,8,9]. Polymers can be designed to encapsulate drugs via covalent bonds to release drugs [9]; (ii) Targeting: polymers can be designed to deliver drugs to specific tissues or cells in the body, increasing drug effectiveness and reducing unwanted side effects. [10,11,12,13]; (iii) Protection: polymers can be used to protect drugs from degradation or elimination in the body, ensuring that the drug reaches its target site in an active form [14,15,16]; (iv) Solubilization: Some drugs are poorly soluble in water, which can limit their effectiveness. Polymers can also be used to solubilize these drugs, allowing them to be delivered more effectively [17,18,19]; (v) Biocompatibility: polymers can be designed to be biocompatible with the body, reducing the risk of adverse reactions and improving patient outcomes [20,21,22]; (vi) Immunogenicity reduction: some polymers can be used to reduce the immunogenicity of drugs, which can be particularly important for biologic drugs such as proteins [23,24]. Overall, the use of polymers in drug delivery offers a wide range of benefits and has led to the development of new and innovative drug delivery systems that are more effective, targeted, and patient-friendly. Artyukhov et al. synthesized biophilic copolymers of various compounds that can self-assemble in water with the formation of polymeric nanoparticles and are suitable for ionic binding of the common anticancer drug doxorubicin for drug delivery application [25]. Accordingly, the copolymers were synthesized by the radical copolymerization of N-vinyl-2-pyrrolidone and acrylic acid using n-octadecyl-mercaptan as the chain transfer agent. According to the obtained data, it was determined that the decrease in the doxorubicin release rate constant is affected by the increase in the concentration of carboxyl groups of acrylic acid residues and the increase in the copolymer concentration. It was also found that this situation is affected more than the composition change. They attributed this to the electrostatic repulsion of doxorubicin cations when the immobilization centers are located in the same chain. Considering all cases, they determined that the kinetics of doxorubicin release fit the equation for reversible first-order reactions.

Polymers also play a crucial role in organ transplantation by providing a variety of functions that help to improve the success of the transplantation process [26,27,28]. Some of the key roles of polymers in organ transplantation include: (i) Immune suppression: polymers can be used to deliver immunosuppressive drugs to the recipient, which helps prevent the immune system from rejecting the transplanted organ [29,30,31]; (ii) Tissue engineering: polymers can be used to create scaffolds for tissue engineering, which can help to regenerate damaged tissues and promote the growth of new organs [32,33,34]; (iii) Anti-bacterial properties: polymers can be designed to have antimicrobial properties, which help to prevent infection and inflammation following transplantation [35,36,37]; (iv) Encapsulation: Polymers can be used to encapsulate islets of Langerhans to protect them from the immune system, while still allowing insulin to be released into the body. This can be used to treat diabetes by transplanting islets into the patient [38,39,40]; (v) Delivery of growth factors: polymers can be utilized to deliver growth factors that promote the growth and regeneration of blood vessels and other tissues, thus helping to improve the success of transplantation [28,41]. The use of polymers in organ transplantation offers a range of benefits that help improve the success of the transplantation process and improve patient outcomes. Ongoing research is exploring new applications of polymers in organ transplantation, including the use of 3D printing to create complex tissue scaffolds and the development of new drug delivery systems that improve the efficacy of immunosuppressive drugs. 

Another active area of research regarding polymers is their design and fabrication for artificial organs, which are engineered devices designed to replace the function of a natural organ [42,43,44]. The use of polymers in artificial organ design offers several advantages including: (i) Biocompatibility: Polymers can be engineered to be biocompatible, which means they are less likely to cause an immune response when implanted into a human body [20,45]. This is crucial for the long-term success of an artificial organ; (ii) Flexibility: Polymers can be designed to acquire flexibility, a crucial property for correctly mimicking the mechanical properties of natural organs. For example, the elasticity of a polymer can be tuned to match that of a natural organ, thereby improving its functioning and reducing the risk of damage [46]; (iii) Customizability: Polymers can be easily fabricated into different shapes and sizes, making them ideal for designing custom artificial organs for individual patients. This can improve the functionality and effectiveness of the artificial organ [47]; (iv) Drug delivery: Polymers can be used to deliver drugs or other therapeutic agents directly to the site of the artificial organ [48]. This can help to prevent infections, reduce inflammation, and promote tissue regeneration; (v) Imaging: Some polymers can be designed to be radio-opaque, which makes them visible on biomedical imaging scans [49]. This is crucial for monitoring the function and health of the artificial organ over time. Ongoing research is exploring new directions for polymer applications in artificial organ design, including the use of 3D printing to create complex and intricate structures and the development of new polymers with unique biological and mechanical properties. 

Polymers can play a crucial role in immune compatibility, which refers to the ability of a material to avoid triggering an immune response when implanted into the body [50,51,52]. In the context of biomedical devices and drug delivery systems, immune compatibility is an important consideration, because an immune response can lead to inflammation, tissue damage, and failure of the device or treatment. Polymers can be engineered to be immune compatible in several ways including: (i) Biocompatibility: Polymers can be designed to be biocompatible, which means that they do not elicit an immune response. This can be achieved by selecting polymers that are non-toxic and non-inflammatory, and by minimizing their interaction with proteins and immune cells [45]; (ii) Surface modification: The surface of a polymer can be modified to reduce its interaction with immune cells and proteins. For example, the surface can be coated with a layer of a biocompatible material, such as a protein or glycan, to reduce its recognition by the immune system [53,54]; (iii) Immunomodulatory properties: Polymers can be designed so that they possess immunomodulatory properties, which means that they can influence the immune response in a beneficial way [55]. For example, some polymers can promote the production of anti-inflammatory cytokines, which can help to reduce inflammation and improve healing [56,57]; (iv) Drug delivery: Polymers, for example, can be used to deliver immunosuppressive drugs to prevent the rejection of transplanted organs or to deliver anti-inflammatory drugs to reduce inflammation [58]. The ability to engineer polymers for immune compatibility is an important consideration in the design of biomedical devices and drug delivery systems. By minimizing the immune response to these materials, it is possible to improve their effectiveness and reduce the risk of complications for patients. 

Various biodegradable polymers have been synthesized, and their structure–property relationships have been studied [59]. Biodegradable polymers are now in demand with increasing applications in biotechnology and biomedical engineering such as drug delivery and tissue engineering (see above). To fulfill these new demands, biodegradable polymers have been found to be potential candidates owing to their characteristic ability to manipulate their physicomechanical properties. This can be achieved by regulating the nature and ratio of the starting material used for polymer synthesis. There are several available polymer classes that demonstrate the potential to be used as biodegradable biomaterials for health and medicine applications. However, perhaps the most promising are biodegradable polyurethanes [60,61,62,63]. Polyurethane contains urethane linkages within the polymer chains. The urethane linkage is equivalent to the carbamate linkage in organic chemistry. The capability of a polyurethane structure to incorporate other functional groups into the polymer network makes it more versatile as compared to other available biomaterials. Polyurethanes can be designed to have specific features, such as hardness, abrasion, chemical resistance, elastic and mechanical properties, and other health and medicine related properties such as blood and tissue compatibility [64]. Biocompatibility and biodegradability are not the only properties that encourage cell growth and proliferation. An ideal degradable biomaterial will have biological and mechanical properties compliant with a suitable degradation mechanism and the ability to be easily fabricated. Polyurethane offers various advantages in designing biomaterials that fulfill these demands [63,65]. All in all, the flexibility of polyurethane synthesis—along with its processing and biofriendly characteristics—has made it a preferred choice over other available synthetic polymers for health and medicinal applications.

Well-known examples of intrinsically disordered protein polymers are polymers embracing intrinsically disordered regions derived from elastomeric proteins, including resilins, elastins, proteins from spider silk, fibrillin, titin, and gluten [66,67]. Elastin-like polypeptides (ELPs), a class of thermo-responsive bioengineered proteins, have emerged as a remarkable model of IDP owing to their low sequence complexity and the similarity of their biophysical characteristics to those of IDPs. The molecular structure of ELPs is composed of repeat units of a Val-Pro-Gly-X-Gly pentapeptide sequence in which X is the guest residue. It can be any amino acid residue except proline [68]. For example, Acosta et al. have synthesized antimicrobial peptides (AMPs) that possess an amphipathic nature with antimicrobial and immunomodulatory properties and also the capacity to self-assemble into supramolecular nanostructures [69]. For this purpose, they linked AMPs to an elastin-like recombinamer (ELR) in their synthesis. They exploited the ability of these AMPs and ELRs to self-assemble to develop supramolecular nanostructures by way of a dual-assembly process. They found that AMPs trigger nanofiber formation, whereas ELRs enable assembly into fibrillar aggregates. Quiroz and Chilkoti presented sequence heuristic guiding principles for encoding LCST and UCST (lower and upper critical solution temperature) phase behavior in intrinsically disordered protein polymers [1]. In summary, they present new tools for studying phase behavior in biology or for exploiting phase transition in diverse fields including materials science. The fusion of ELP genes encoding segments of different pentapeptide sequences permits the synthesis of ELP block copolymers. Conticello and coworkers were the first to synthesize ELP diblock copolymers composed of hydrophilic VPGEG-(IPGAG)4 and hydrophobic VPGFG-(IPGVG)4 blocks [70]. The distinct sequences of each block allowed them to retain their independent thermal response. The reversible temperature-dependent assembly of these nanoscale structures was verified by performing differential scanning calorimetry (DSC) and dynamic light scattering (DLS). Transmission electron microscopy (TEM) images confirmed the spherical and, in some cases, cylindrical morphology of these particles. An ELP triblock was synthesized by capping a central hydrophilic domain with VPAVG-(IPAVG)4 hydrophobic blocks. These triblocks formed extended networks of micellar nanoparticles connected by cross-links composed of the central hydrophilic domain [71]. The precise control over the size and the stimulus-responsive characteristics of ELPs provides a useful platform for the design of macromolecular carriers for drug delivery. Furthermore, genetic engineering of ELPs permits the incorporation of targeting peptides, such as cell-penetrating domains. ELPs are attractive cancer drug carriers because their biocompatibility, synthesis, and stimulus responsiveness provide tunable properties that can be optimized for a specific drug. Chilkoti and co-workers synthesized ELPs and monitored the distribution and accumulation of fluorescently tagged conjugates of their ELPs in tumors [72]. The temperature-triggered aggregation of an ELP in heated tumors resulted in a twofold increase in accumulation in comparison with temperature insensitive ELPs in tumors that were not heated. Their in vivo results were paralleled in vitro with an enhanced uptake of thermally responsive ELP [73]. Furthermore, the inclusion of an N-terminal lysine on the ELP chain allowed the conjugation of doxorubicin (Dox), a commonly used chemotherapeutic, through a pH-sensitive hydrazone linker. The reaction of ELP with succinimidyl-4-(N-maleimidomethyl)cyclohexane-1-carboxylate (SMCC) functionalized the lysine residue with a reactive maleimide group that could then be reacted with Dox-hydrazone, resulting in an ELP-Dox conjugate. The conjugation of Dox, using this method, at its 13-keto position, permitted the retention of the cytotoxicity of Dox. Following endosomal uptake, the pH-labile hydrazone linker released free drug in the acidic lysosomal compartments. When incubated with FaDu cells, these ELP-Dox conjugates were endocytosed and transported into lysosomes as detected by confocal fluorescence microscopy [74]. A C-terminal with cysteine on the ELP chain was also engineered to conjugate a maleimide derivative of Dox to the ELP through a pH-sensitive hydrazone linker [75]. Moreover, Luo et al. determined that interactions between collagen-like peptides (CLPs) and native collagens on the surface of elastin-b-collagen-like peptides (ELP-CLPs) vesicles could be used to target these vesicles to collagen-containing substrates [76]. For this purpose, they performed retention experiments on vesicles on type II collagen films. They demonstrated the extent to which they use ELP-CLP vesicles as an aid to heat-sensitive drug release to target the collagen-containing matrices they have made. Here, it provides the sustained release of a clinically relevant cargo transport in 3- and 6-week periods. Additionally, a heat-sensitive burst release was observed as the vesicles decomposed above the CLP opening temperature, thus using the hyperthermia process to trigger the encapsulated release from ELP-CLPs using appropriate lengths.

ELPs are thermally sensitive and have a temperature-reversible phase transition [77]. Thus, the ELP molecules self-assemble with a temperature increase above a hydrophobically characteristic transition temperature (Tt) to shape an extremely viscous liquid (coacervate) [77]. The ability of ELPs to be designed to approximate the viscoelastic properties of natural elastin upon cross-linking, as well as their being biocompatible, biodegradable, and non-immunogenic, has increased their use in tissue engineering applications in recent years. Due to the temperature sensitivity of ELPs, they can be used for tissue engineering applications where biomaterials are required that can be injected and somehow triggered to form a solid matrix after the defect is filled. ELPs can also be designed to obtain a scaffold with mechanical stability after cross-linking. In addition, this scaffold can be developed to be mixed with a biocompatible cross-linker, which is also triggered by temperature or another stimulus in the environment. Modifications in mechanical, swelling, degradation, and cross-linking properties can be made by means of block copolymers made with ELPs by alternative groups containing hydrophobic, hydrophilic, cross-linking, and cell-recognition sequences. Additionally to all these properties, ELPs are easily synthesized and easily cross-linked to form foams, gels, and fibers for use in tissue engineering applications.

Despite the demonstrated versatility of the elastins and the longstanding knowledge of the unique properties of related resilin, the development of resilin-based materials was hindered due to challenges related to batch-to-batch variability, isolation, and limited supply. Ardell and Andersen report a 620-amino acid sequence similar to resilin’s sequence found in a variety of insect proteins [78]. The highly conserved repeat sequences were consistent with the precursor encoding for resilin. Next, Elvin et al. cloned and expressed the first exon of this sequence as the first resilin-like protein (rec1-resilin) in Escherichia coli [79]. Circular dichroism and computational studies demonstrate that the structure of rec1-resilin is 85–95% random coil [80]. Biomaterials with hybrid mechanical properties based on resilin-like peptides and poly(ethylene glycol) (PEG) conserve the inherent properties defined for resilin-like peptide systems [81]. Upon cross-linking, the evolving microstructure of the emulsion can be captured in a hydrogel that enables the localization of mechanically distinct resilin-like peptide-rich domains. The dynamic nature of the associated phase separation allows temporal control over the microdomain dimensions and opportunities to guide cell localization around mechanically relevant regions of adequate dimensions, predefining cell distribution, which may afford advantages in guiding tissue engineering. Okesola et al. reported a covalent co-assembly strategy based on peptide amphiphile (Pas) and resilin-like peptides for gaining control over the hierarchical assembly of RLPs, thereby obtaining a hybrid material with good mechanical properties [82]. They aimed to integrate the functionality of covalent interactions with the complexity provided by multi-component self-assembly. For this purpose, covalent interactions in the presence of a thiol-ene photoclick reaction were achieved by the assembly of sulfhydryl group functionalized PA and acrylamide functionalized RLPs together. Dynamic amplitude sweep rheology was applied to examine the synergistic effect of recombination and covalent interaction on mechanical properties. Here, co-assembled RLP-PA hydrogels (thiol-ene photoclick) exhibited a G’ value of ∼4.5 kPa, indicating that a covalent interaction predominated in the co-assembly system, whereas a value of 1.6 kPa was recorded in hydrogels without thiol-ene photoclick-based covalent interactions. They also showed that the high flexibility of the combined hydrogels may indicate that the chain mobility of RLP is retained in the hydrogels. They reported that all these properties could open up opportunities for applications such as the fabrication of scaffolds for tissue formation or sustained drug release systems.

Using polyurethanes, one can obtain more complex structures and good mechanical properties. Recently, we introduced a new class of polymers (synthetic intrinsically disordered polymers, sIDPs) for soft robotics applications based on polyurethane [83]. Herein, we present new synthesis mechanisms for synthetic intrinsically disordered polymers (sIDPs) to be used in health and medicine using polyurethane. These sIDPs are bioinspired by intrinsically disordered proteins that have multiple biological functions and are at the center of various maladies including neurodegeneration, where they form specific aggregates known as amyloid-like fibrils [84,85]. Strategies for the characterization of sIDPs and machining the bulk polymer structures into filaments for 3D printing purposes are discussed herein. These strategies and mechanisms presented in this article pave the way for the provision of specific polyurethane-based sIDPs, which are currently in great demand in various application areas in health and medicine.

## 2. Intrinsically Disordered Synthetic Polymers

### 2.1. Biomedical Applications

Intrinsically disordered proteins (IDPs) play a number of unique and important roles in various biological processes due to their inherent flexibility and ability to undergo con-formational changes in response to environmental cues (Table 1) [23,24,25]. These properties also make IDPs well suited for a variety of biomedical applications [26,27,28,29,30,31,32,33], including: (i) Self-assembly: IDPs can self-assemble into a variety of nanostructures, such as micelles, nanoparticles, and hydrogels, which can be used for drug delivery; (ii) Controlled release: the ability of IDPs to undergo conformational changes can also allow for the controlled release of drugs; (iii) Targeting: IDPs can be designed to recognize specific biological targets, such as cells and proteins, through binding interactions. This can allow for targeted drug delivery to the specific tissues or cells in the body, which can improve drug efficacy and reduce side effects; (iv) Stabilization: IDPs can be used to stabilize proteins or other therapeutics, which can improve their pharmacokinetics and bioavailability; (v) Biocompatibility: IDPs are generally biocompatible and biodegradable, which can reduce the risk of toxicity and improve patient outcomes; (vi) Immunogenicity reduction: IDPs can be used to reduce the immunogenicity of drugs, which can be particularly important for biologic drugs such as proteins; (v) Immune suppression: IDPs can be used to deliver immunosuppressive drugs to the recipient, similar to polymers, which helps prevent the immune system from rejecting the transplanted organ. IDPs can be specifically designed to target certain immune cells and deliver immunosuppressive drugs directly to them, potentially reducing the amount of drugs required; (vi) Tissue engineering: IDPs can be used in the creation of tissue scaffolds, similar to polymers, to support the growth and regeneration of new tissues following transplantation; (vii) Anti-inflammatory properties: IDPs have been shown to have anti-inflammatory properties, which can be beneficial in reducing the immune response and inflammation associated with transplantation; (viii) Self-assembly: IDPs have the ability to self-assemble into specific structures, which makes them useful for creating complex and organized structures in artificial organs. This can be useful for designing complex tissues or organs, such as pancreas or liver; (ix) Multifunctionality: IDPs can perform multiple functions within a single system. For example, an IDP could act as a scaffold for tissue growth, while also delivering drugs or therapeutic agents to the site of the artificial organ; (x) Dynamic behavior: IDPs are extremely flexible and dynamic, which is useful for mimicking the mechanical properties of natural organs. This can be important for designing artificial organs that require complex movements, such as hearts and/or lungs; (xi) IDPs have the ability to interact with immune cells and modulate the immune response. For instance, IDPs can activate or inhibit immune cells, depending on the desired outcome. This can be useful in the context of immune therapy, where the goal is to activate the immune system to fight cancer or other diseases; (xii) Antimicrobial properties: Some IDPs have antimicrobial properties, which means they can kill bacteria and other pathogens. This can be useful in the context of biomedical devices, in which bacterial infection can be a serious concern. All in all, the unique properties of IDPs make them a promising platform for drug delivery. Ongoing research is exploring their potential for a wide range of therapeutic applications.

Segmented polyurethanes represent a crucial class of synthetic polymers for potential health-related and biomedical applications [86,87]. Using non-toxic soft segment polyols, hard segment chain extenders enable the development of an entirely new family of biodegradable polymers, which may exhibit diverse properties and may be suited for a wide range of applications [88]. However, these need to be biocompatible and biodegradable. To enhance the degradation process, hydrolysable linkages may be inserted via chain extenders, leading to the degradable hard segments, which are usually the segments that degrade very slowly in polyurethane [89]. This approach may be less common, but amino acid or peptide-based chain extenders with hydrolysable ester linkages were synthesized and incorporated into polyurethane [90]. The idea of incorporating amino acids in the form of a chain extender has various benefits: (i) Non-toxic products would be released upon polymer degradation; (ii) Enzyme-mediated degradation can be tailored into the polymer with regard to the known amino acid based enzyme profile at the site of application, (iii) The side chain functional group of different amino acid residues can be used to generate a pendant group on the polymer backbone. These pendant groups can be used as reactive sites, such as sites engaged in drug carrying. Different amino acid residues can be added to the chain extenders to develop polyurethanes for drug delivery, tissue engineering, artificial organ design, and immune compatibility.

Intrinsically disordered proteins (IDPs) lack stable three-dimensional structures, possess high flexibility, and are prone to self-organization. There are seven disorder-promoting (or structure-breaking) amino acid residues: these are glycine, lysine, histidine, proline, arginine, glutamic acid, and glutamine [83]. From a biological perspective, IDPs can be fully unstructured or partially structured and include random coil, molten globule-like conformers, or flexible linkers in large multidomain proteins [91,92,93]. IDPs participate in weak multivalent interactions, which are dynamic, highly cooperative, and easily amenable to minute environmental changes that make them important in signaling [94]. Various IDPs can adopt at least a partially ordered structure after binding to partners including small molecules [95]. Overall, IDPs differ from structured/ordered proteins in various aspects and tend to possess specific functions, structures, sequences, interactions, regulation, and evolution. Approximately 20 years ago, it became clear that IDPs are common among disease-related proteins. We have been actively studying the structure-function relationships of IDPs and the impacts of specific amino acid residues on the structure-function relationships of IDPs for more than 25 years.

Here, we propose to bio-mimic intrinsically disordered proteins that gain a function based on the ligand binding and could have various multifunctional properties based on ligand binding coordination chemistry variations [96,97]. They can also self-assemble in large specific complexes that make them attractive targets for self-healing [85]. Self-healing polymers are, for example, important in drug delivery because they have the ability to repair themselves when damaged, which can improve the longevity and stability of drug delivery vehicles. Such polymers are designed to respond to specific stimuli, such as changes in pH or temperature, and can undergo reversible changes in their chemical structure in response to these stimuli [98,99]. In drug delivery, self-healing polymers can be used to create drug delivery vehicles that are more stable and can better protect drugs from degradation and premature release. In addition, self-healing polymers can improve the targeted delivery of drugs to specific cells or tissues. By using polymers that respond to certain stimuli found in certain parts of the body, drug delivery vehicles can be designed to release drugs only in those areas. This can improve the efficacy of drugs while reducing side effects.

Artificial organs are designed to replace or assist the function of damaged or diseases organs in the body. However, artificial organs are often subject to wear and tear over time, which can lead to failure or the need for replacement. Self-healing polymers can help to address this issue by allowing artificial organs to repair themselves when they become damaged, extending their lifespan and reducing the need for replacement [100,101]. Self-healing polymers can also improve the biocompatibility of artificial organs, reducing the risk of rejection or other adverse reactions by the body [102,103]. This is particularly important in the case of organ transplantation, where the body’s immune system can sometimes identify the transplanted organ as foreign and attack it. Moreover, self-healing polymers can be used to create drug delivery systems within artificial organs [104,105]. This can help to deliver drugs or other therapeutic agents directly to the side of the organ, promoting healing and reducing the risk of rejection. 

When a biomedical device or implant is introduced into the body, it can sometimes trigger an immune response, which can lead to rejection or inflammatory reactions. Self-healing polymers can help to reduce the risk by improving the compatibility of the device with the body’s cells and tissues [106,107]. The synthetic intrinsically disordered polymers to be produced by our group will be mostly used in biomedical application areas.

### 2.2. Synthesis of Intrinsically Disordered Polymers for Biomedical Applications

Two different approaches have been developed for obtaining synthetic intrinsically disordered polymers for biomedical applications. Biocompatibility and self-healing are taken into account in our approaches. A sequence determines the function of a protein. Seven structure-breaking amino acid residues can be used in various sequential orders for gaining different functionalities [83]. Here, we provide as an example the oligomer sequence Gly-His-Lys-Pro-Arg-Glu-Gln. 

#### 2.2.1. Modification of the Chain Extender with a Structure-Breaking Peptide Oligomer

The first of these two approaches involves the modification of the chain extender with a structure-breaking peptide. First, a complex is formed by the example structure-breaking peptide sequence reacting with monobromo triethylene glycol (Figure 1). 

This complex acts as a new type of chain extender. Modification takes place in one step. The structure-breaking peptide is mixed at a constant mixing speed until it is completely dissolved into distilled water at 50 °C. After complete dissolution, monobrome triethylene glycol is added, and the reaction continues for 4 h. The resulting complex is dialyzed against ultrapure water for 3 days using a dialysis membrane with appropriate pore size for removing impurities from the medium. The protection reaction is then carried out using tert-butyl hydrogen carbonate (BOC) for protection of the amine groups. 

#### 2.2.2. Synthesis of the Prepolymer and Synthesis of Intrinsically Disordered Polymer

After the chain extender is obtained, the next step involves the preparation of the prepolymer. As we have mentioned, a controllable reaction system is very crucial for the production of polyurethanes. For this purpose, the two-step polyaddition reaction is preferred. However, when the goal is to obtain a biodegradable polymer, it must be ensured that the polymer does not leave a toxic effect after its degradation. For obtaining biocompatible and self-healing intrinsically disordered polymers, we chose the non-toxic L-lysine ethyl ester diisocyanate as the hard segment and poly(ethylene glycol) as the soft segment.

First, L-lysine ethyl ester diisocyanate (2 eq.) and PEG (1 eq.) are mixed into a DMF solution at 90 °C under constant stirring at 400 rpm for 2 h (Figure 2). Immediately afterwards, the new chain extender (Complex 1) is added to the solution with the prepolymer at a ratio of 1 molar in an inert atmosphere under continuous mixing (Figure 3). After this addition is complete, the reaction continues for 3 h at constant temperature. The resulting viscous solution is dried in a vacuum oven for 36 h. The final product is obtained by extracting the polymer in a chloroform solution. All reactions are carried out under an inert atmosphere.

#### 2.2.3. Synthesis of Intrinsically Disordered Polymers

A second approach that we designed for the production of biodegradable and self-healing intrinsically disordered polymers involves modification of the synthetic intrinsically disordered polymers that we recently designed for soft robotics applications [83]. For this purpose, the polymerization steps are performed in the presence of chain extenders with protecting groups that we designed in our previous study (Figure 4) [83]. Here, chain extenders suitable for the production of biocompatible polymers were selected. Exemplary polymerization reaction mechanism is demonstrated in the presence of a chain extender with the functional group of lysine((4-aminobutoxy)butane) (Figure 5). After obtaining the disordered polymer, the first deprotection reaction is performed, and the impurities in the medium are removed by washing with chloroform (Figure 6).

The peptide is then added to the intrinsically disordered polyurethane chain dissolved in DMF, and the reaction takes place at room temperature under constant stirring for 4 h (Figure 5). 

The resulting final polymer is extracted for the removal of impurities in the medium. Figure 6 shows some other biodegradable and self-healing intrinsically disordered polymers that can be obtained using similar reaction mechanisms. 

### 2.3. Characterization and Production of New Class of Bio-Mimicked Intrinsically Disordered Polymers

Rheological analyses are necessary to ascertain the flow characteristics of polymers before they are created. Rheological analysis is conducted utilizing a rotational rheometer equipped with a 25 mm diameter parallel plate geometry to understand the flow properties of the polymers. A stress-strain sweep test is recorded at a constant strain of 1% in a frequency range of 0.1–100 rad/s at room and significant temperatures to characterize the samples. To examine self-healing characteristics, the measurement of the recovery characteristics by utilizing rheology analysis is crucial. For this purpose, an oscillation-time sweep test can be performed with strains varying from 250% to 0.5%.

When the materials are in the proper state to be extruded, the polymers are granulated to a size of 4 mm ± 1 mm in diameter at room temperature. Before loading them into the extruder, pre-drying is needed. This procedure is performed in 2 to 4 h in air circulation ovens that operate at 90–100 °C. To obtain the filaments, a double-screw extruder through a cylindrical nozzle (ø 2.9 mm) is needed, utilizing a certain pressure at determined T_m_ values of the polymers. The filament diameter is controlled by an electronic caliper to obtain the desired filament size.

It is also of interest to determine the mechanical properties after the healing process. Therefore, the materials are cut using a razor and then rejoined and held under dark conditions for 24 h. The above-described mechanical testing is applied to specimens after healing. Dynamic thermomechanical analysis (DMA) tests are necessary to investigate the thermomechanical behavior of 3D-printed polymers. These will be conducted on a DMA analyzer by decreasing the temperature. According to the peak of tanθ, the specimens’ glass transition temperature (T_g_) will be identified.

The drug release rate can be determined by measuring the amount of drug released over time [108]. This can be effected by using various analytical techniques, such as UV-Vis spectroscopy, high-performance liquid chromatography (HPLC), or mass spectrometry.

To study the drug release behavior of self-healing polymers, a drug is determined and encapsulated into the polymer. For this purpose, the total drug content is calculated as a function of the sample weight and the predicted weight ratio. The drug-encapsulated self-healing polymers are then immersed into phosphate buffered saline (PBS) solution at 37 °C to determine drug-release behavior. A certain amount of solution is taken from the PBS medium at specified time intervals, and then the same amount of fresh PBS is added to the system. A certain amount of solution is measured with a UV-Vis spectrophotometer at the characteristic absorption value of the selected drug. To make the necessary calculations, the calibration curve is first determined. In this context, an absorbance-concentration calibration curve is first calculated, ranging from 5 to 100 ppm. The calibration curve is y = mx + n, where y represents the absorbance value of the solution at the characteristic absorption value of the drug, and x represents the drug concentration (ppm). Cumulative drug release curves are plotted using this determined calibration curve. Experiments are performed in triplicate.

To determine the amount of drug encapsulated, drug-loaded self-healing polymers are dipped into PBS and incubated for 72 h. The absorbance of the supernatant solution is measured in the range of the specific absorbance value of the drug determined by the UV-Vis spectrophotometer. Drug loading efficiency is determined using the standard curve of drug release and is calculated from the following equation:$$Drug\;Loading\;Efficiency\left(\%\right)=\frac{Amount\;of\;maximum\;Drug\;release}{Initial\;amount\;of\;Drug\;containing\;beads}\times 100$$

Drug release from a polymer matrix usually implies water penetration in the matrix, hydration, swelling, diffusion of the dissolved drug, and the erosion of the gelatinous layer [109]. It is crucial to mention that the release mechanism of a drug depends on the drug dose, investigation of the solution pH, and the nature of the polymer and drug used [2]. The amount of drug released can be correlated with the degree of swelling of the polymer matrix [110].

Cytotoxicity can occur when cells are adversely affected by chemical substances or the physical properties of the environment [111]. To understand the biocompatibility of a biomaterial, it is necessary to determine its toxic or non-toxic effect on cells. A cytotoxicity test is used to assess the potential toxicity of self-healing polymers on cells. For this purpose, the self-healing polymer is typically sterilized and prepared in a form that is suitable for implantation, such as disks or fibers.

An indirect MTT test is performed to assess the cytotoxicity of the self-healing polymer. A quantity of 2 mL of MEM Alpha medium is added to completely dry samples and incubated at 37 °C with 5% CO_2_ for 24 h. The medium containing the incubated self-healing polymer extracts is sterilized with a 0.22 µm syringe filter. Osteoblasts are seeded at 20 × 104 cells/well with a given amount of MEM Alpha per well in a 48-well plate. It is then incubated at 37 °C with 5% CO_2_ for 24 h. The osteoblast cells are then replaced with a culture medium containing biopolymer extracts and incubated for 24 h. A certain amount per well of MEM Alpha and MTT solution prepared in PBS is added and incubated for an additional 4 h for formazan crystal formation. The solution is then removed, and a certain amount of DMSO is added per well to dissolve these crystals. The formazan solution, which is incubated for an additional 20 min, is read at 570 nm using a microplate reader. Wells with MEM Alpha medium are prepared as a control sample.

Intrinsically disordered polymers can be measured to assess their biodegradability [112]. These bio-mimicked polymers are designed to mimic intrinsically disordered proteins that can be degraded by enzymes in the body. Enzymatic degradation assays can be used to assess the biodegradation extent of bio-mimicked intrinsically disordered polymers. 

To describe the degradation study in more detail, a degradation medium containing collagenase type II is prepared in DPBS to determine the enzymatic degradation of the self-healing polymers. For in vitro degradation experiments, the self-healing polymers are weighed dry and incubated at 37 °C for 1, 3, 5, 7, 10, 24, and 48 h in degradation solution containing collagenase type II. Afterwards, the polymers taken from the degradation medium are dried in a vacuum oven for 24 h and weighed again. For long-term degradation studies, DPBS is used as the degradation medium. Samples incubated here at 37 °C are removed from the environment at the end of the 21st day and dried. Dry samples are then weighed. The degradation rate is calculated according to the equation given below. Four repetitions are run for each sample. In addition, samples taken from the degradation medium and weighed are examined by FTIR analysis to determine their chemical structure.
$$Degradation(\%)=\frac{\left(Wd-Wi\right)}{Wi}\times 100$$
where *W_i_* is the initial completely dry weight of the samples, and *W_d_* is the dry weight after incubation at a particular data point.

## 3. Conclusions

Here, we present new chemical reaction mechanisms and experimental designs for the studies of polyurethane polymers that bio-mimic intrinsically disordered proteins. These are a new class of polymers that are flexible and biodegradable and possess self-healing capacities. These polymers can be manufactured by polymer and material industries for various purposes, including biomedical applications such as drug delivery, artificial organ design, organ transplantation, and immune compatibility. 

## Data Availability

All data are presented.

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
