# Peer review of "Intrinsically Disordered Synthetic Polymers in Biomedical Applications"

_polymers, 2023, doi:10.3390/polym15102406_

Round 1
Reviewer 1 Report
The topic of the manuscript is interesting. However, there are some issues that need to be addressed and questions that need to be answered. 1. The authors have already published an article discussing the novelty of the approach and mechanisms (ref.85 in the manuscript). It would be good, if the authors indicate the novelty of this article more clearly. 2. It is also necessary to discuss in more detail what structures of polymers can be and what are in the literature for use in medicine, and why (on what basis) the authors chose those that are discussed in this work. The work still claims the category 'review'.Author Response
Thank you very much!
Ref 85 considers non-biodegradable polymers and this manuscript is about biodegradable polymers useful in biomedical application areas. We added elastins and resilin-like peptides and their definitions and properties in the introduction part. Ref. 85 introduced new synthesis mechanisms but it is a review article. We added elastin and resilin-like polymers and enhanced the review manuscript.
Reviewer 2 Report
Undoubtedly, this REVIEW is exciting, but unfortunately, the idea of this REVIEW has not been described and explained thoroughly in the manuscript. Therefore, I believe that this manuscript needs a significant revision.
My comments are listed below.
1. The introduction is tedious and quite confusing, repeating the same sentences. Comparisons between IDPs and synthetic polymers could be conducted more clearly, pointing out each group's key advantages and disadvantages in two separate paragraphs. I will also suggest a table where the two groups will be compared.
2. In the Section Biomedical Applications, lines 273-296 are irrelevant to the paragraph.
3. If I understood well, the central part of the text is supposed to be about the examples of intrinsically disordered polymers. You have just given the experimental details (unnecessary) of a synthesis. Are there any other examples in the literature? It would help to compare your strategy with similar techniques presented in the literature.
4. Parts 2.3 to 2.8 of the manuscript should be merged into one named characterization and application of intrinsically disordered polymers. There are a lot of unnecessary details.
5. Overall, I will not consider the current manuscript in this form as a REVIEW.
Author Response
Answers:
- Thank you very much! We rewrote the introduction part of the manuscript.
- Thank you very much. Lines 273-296 about self-healing were removed.
- We included also elastin and resilin-like polymers which are considered as intrinsically disordered protein polymers.
- These subsection were merged together and the kinetic description in drug release studies were removed.
- We hope that we made the necessary changes even though this review manuscript also introduces new synthesis mechanisms similar to the review article https://www.mdpi.com/2073-4360/15/3/763
Reviewer 3 Report
The review is aimed at considering the synthesis and properties of polyurethanes for biomedical applications. The review requires significant expansion in relation to the cited literature. Among the cited works, there is a significant number of the authors' own works, which gives the review additional value. However, there are some points that require correction and clarification:
1) What specific types (from chemical standpoint) of polymers do the authors refer to as internally disordered synthetic polymers? It is not clear by what criteria the authors selected the material? The listed functions of polymers as drug carriers are also not exhaustive. This review touches upon this issue in a too general way, without analyzing any structural criteria and without referring to classes of polymers (on the basis of their chemical structure).
2) Why isn't enough attention paid to biphilic polymers as drug delivery systems?
3) In the introduction, it is required to outline the time period for which the literature is being analyzed and clearly identify the subject of the review.
4) On page 1, the ability of the polymers to solubilize drugs is noted, but the ability to coordinate (https://doi.org/10.1002/pi.6073) or covalently bind is not noted.
The volume and breadth of coverage of the literature should adequately correspond to the title of the manuscript. Therefore, either the review requires expansion or concentration on specific types of synthetic polymers.
Author Response
- The new class of polymers is based on disorder-promoting sequences and polyurethane-different than existing studies. However, we included also the literature review about elastin and resilin-like polymers which are intrinsically disordered as well.
-
Thank you very much! The following paragraph is now included in the revised manuscript:
“Artyukhov et al. synthesized biphilic copolymers of various compounds that can self-assemble in water with the formation of polymeric nanoparticles and are suitable for ionic binding of the common anticancer drug doxorubicin for drug delivery application. Accordingly, the copolymers were synthesized by radical copolymerization of N-vinyl-2-pyrrolidone and acrylic acid using n-octadecyl-mercaptan as the chain transfer agent. According to the obtained data, it was determined that the decrease in Doxorubicin release rate constant was affected by the increase in the concentration of carboxyl groups of acrylic acid residues and the increase in the copolymer concentration. It was also found that this situation was affected more than the composition change. They attributed this to the electrostatic repulsion of doxorubicin cations when the immobilization centers were located in the same chain. Considering all cases, they determined that the kinetics of doxorubicin release fit the equation for reversible first-order reactions.” - Elastin and resilin-like peptides and their literature has been now included as well
- This is now included.
- We expanded the manuscript.
Round 2
Reviewer 1 Report
The manuscript has been seriously revised and can be accepted for publication.
Author Response
Thank you very much.
Reviewer 2 Report
Although the manuscript has been improved, the introduction still is confusing. A table with a clear comparison between IDPs and synthetic polymers would help clarify the introduction.
The characterization part of intrinsically disordered polymers contains a lot of unnecessary details that should be removed. [You can remove this part altogether]
Author Response
We added a Table that lists the properties of IDPs and disordered polymers. Additionally, we shortened the characterization part. Thank you!
Reviewer 3 Report
The review is corrected and meets the highest standards.
Author Response
Thank you very much!
Round 3
Reviewer 2 Report
I do not have any other comments/suggestions.